# Field assessment of the operating procedures of a semi-quantitative G6PD Biosensor to improve repeatability of routine testing

Arkasha Sadhewa[1‡]*, Alina Chaudhary[2‡], Lydia V. Panggalo[3], Angela Rumaseb[1], Nabaraj Adhikari[2], Sanjib Adhikari[2], Komal Raj Rijal[2], Megha Raj Banjara[2], Ric N. Price[1,4,5], Kamala Thriemer[1], Prakash Ghimire[2], Benedikt Ley[1‡], Ari Winasti Satyagraha[3,6‡]

**1** Global and Tropical Health Division, Menzies School of Health Research and Charles Darwin University, Darwin, Australia, **2** Central Department of Microbiology, Tribhuvan University, Kirtipur, Kathmandu, Nepal, **3** EXEINS Health Initiative, Jakarta, Indonesia, **4** Mahidol-Oxford Tropical Medicine Research Unit (MORU), Faculty of Tropical Medicine, Mahidol University, Bangkok, Thailand, **5** Centre for Tropical Medicine and Global Health, Nuffield Department of Clinical Medicine, University of Oxford, Oxford, United Kingdom, **6** Eijkman Research Center for Molecular Biology, National Research and Innovation Agency, Cibinong, Indonesia

‡ AS and AC shared first authorship on this work. BL and AWS shared last authorship on this work.
* arkasha.sadhewa@menzies.edu.au

**Data Availability Statement:** All relevant data are within the manuscript and its Supporting Information files.

## Abstract

In remote communities, diagnosis of G6PD deficiency is challenging. We assessed the impact of modified test procedures and delayed testing for the point-of-care diagnostic STANDARD G6PD (SDBiosensor, RoK), and evaluated recommended cut-offs. We tested capillary blood from fingerpricks (Standard Method) and a microtainer (BD, USA; Method 1), venous blood from a vacutainer (BD, USA; Method 2), varied sample application methods (Methods 3), and used micropipettes rather than the test's single-use pipette (Method 4). Repeatability was assessed by comparing median differences between paired measurements. All methods were tested 20 times under laboratory conditions on three volunteers. The Standard Method and the method with best repeatability were tested in Indonesia and Nepal. In Indonesia 60 participants were tested in duplicate by both methods, in Nepal 120 participants were tested in duplicate by either method. The adjusted male median (AMM) of the Biosensor Standard Method readings was defined as 100% activity. In Indonesia, the difference between paired readings of the Standard and modified methods was compared to assess the impact of delayed testing. In the pilot study repeatability didn't differ significantly (p = 0.381); Method 3 showed lowest variability. One Nepalese participant had <30% activity, one Indonesian and 10 Nepalese participants had intermediate activity ($\geq$30% to <70% activity). Repeatability didn't differ significantly in Indonesia (Standard: 0.2U/gHb [IQR: 0.1–0.4]; Method 3: 0.3U/gHb [IQR: 0.1–0.5]; p = 0.425) or Nepal (Standard: 0.4U/gHb [IQR: 0.2–0.6]; Method 3: 0.3U/gHb [IQR: 0.1–0.6]; p = 0.330). Median G6PD measurements by Method 3 were 0.4U/gHb (IQR: -0.2 to 0.7, p = 0.005) higher after a 5-hour delay compared to the Standard Method. The definition of 100% activity by the Standard Method matched the manufacturer-recommended cut-off for 70% activity. We couldn't improve repeatability. Delays of up to 5 hours didn't result in a clinically relevant difference in

**Funding:** This work has received grant funding from the Australia-Indonesia Institute of the Department of Foreign Affairs and Trade of Australia (AII202100069) awarded to BL. AS is supported by Charles Darwin International PhD Scholarships (CDIPS). Publication costs were funded in part by the Division of Teaching of the Menzies School of Health Research

**Competing interests:** The authors have declared that no competing interests exist.

measured G6PD activity. The manufacturer's recommended cut-off for intermediate deficiency is conservative.

## Introduction

*Plasmodium vivax* (*P. vivax*) causes 4.9 to 14.3 million clinical episodes annually [1, 2] and has become the dominant *Plasmodium* species outside of sub-Saharan Africa. Globally, more than 3.3 billion people are at risk of infection with *P. vivax* [1]. In contrast to most human Plasmodium species, *P. vivax* and *P. ovale* form dormant liver stages (hypnozoites), that can reactivate weeks to months after the initial infection causing recurrent episodes of malaria (relapses) [3, 4]. The risk and frequency of relapses vary depending on the strain [3], with those in equatorial regions relapsing more frequently, associated with a significant health and economic burden [5, 6].

Radical cure of *P. vivax* ensures the clearance of both the asexual blood-stage parasites that cause febrile illness (schizontocidal treatment) and clearance of dormant liver stages (hypnozoites) that can cause relapsing infections. Since schizontocidal antimalarial drugs have no activity against hypnozoites, radical cure requires a combination of schizontocidal and hypnozoitocidal agents. Depending on geographic location, chloroquine (CQ) or artemisinin combination therapies (ACT) are used as schizontocidal treatment [7]. The only currently licensed hypnozoitocides are the 8-aminoquinolines (8AQs) primaquine (PQ) and tafenoquine (TQ) [8, 9]. Although well tolerated in most recipients, 8AQs can induce severe haemolysis in individuals with glucose-6-phosphate dehydrogenase (G6PD) deficiency, a common enzymopathy found in malaria endemic areas [10].

G6PD deficiency (G6PDd) is caused by mutations in the *G6PD* gene, located on the X-chromosome. It affects between 400 to 500 million people worldwide and impairs the red blood cells' (RBC) ability to regulate cellular redox potential whilst ensuring their survival against oxidative stressors [11]. Males have one X-chromosome and are either hemizygous G6PD normal (the vast majority having >70% G6PD activity) or G6PD deficient (the majority having <30% G6PD activity) [12, 13]. Females have two copies of the gene and can be G6PD homozygous normal, G6PD heterozygous, or G6PD homozygous deficient [14]. In heterozygous females, G6PD deficient and G6PD normal RBC populations exist with varying proportions [15]. G6PD activities of heterozygous females range from close to 0% to almost normal activity with the majority of females having intermediate activities between 30% to 70% of normal activity [12].

In the presence of strong oxidants, such as 8AQs, G6PD deficient RBCs can haemolyse, resulting in a severe drop in haemoglobin (Hb) concentrations leading to haemodynamic instability and potentially life-threatening acute kidney injury [16]. The World Health Organization (WHO) therefore recommends that all patients are tested for G6PDd prior to administration of either PQ or TQ [7, 17]. Diagnosing individuals with intermediate G6PD activity is also recommended to exclude these patients from administration of TQ [18] and to guide the use of short-course high-daily dose PQ regimens [19, 20]. Diagnosing intermediate deficiency is currently only possible using quantitative testing, due to the low inherent diagnostic thresholds of available qualitative tests [17].

Over the last few years several handheld devices (biosensors) have been introduced to the market capable of measuring G6PD activity quantitatively or semi-quantitatively at point-of-care. The semi-quantitative STANDARD G6PD Test (SD Biosensor, RoK; "Biosensor") has excellent performance under laboratory and field conditions [21–23] and has now been rolled out for routine testing in seven countries [24]. The Biosensor's reproducibility and

repeatability under lab conditions is excellent [25], however anecdotal evidence suggests lower repeatability when the test is applied under field conditions.

As the incidence of malaria declines, the proportion of patients with *P. vivax* infection in remote areas increases [26, 27]. Delivering quantitative G6PD measurements at these remote communities is challenging [24], hence patients diagnosed with *P. vivax* are referred to higher level health care facilities for a G6PD measurement, although the rate of referral is often low [28]. An alternative option to referring patients to the closest health facility, is collection and transport of the patients' blood to the health facility instead. Collected blood can then be tested by Biosensor and the results communicated back to the treating clinician to inform suitable treatment. The stability of G6PD activity of samples stored at 4˚C has been evaluated in the past with spectrophotometry [21] and was found to remain stable over several days, though this has not yet been assessed using a Biosensor.

The objective of this study was to assess whether the repeatability (variation of measurement results if the same sample is tested repeatedly under the same conditions) and reproducibility (variation of measurement results if the same sample is tested repeatedly under different conditions) of the STANDARD G6PD Test (Biosensor) can be improved by modifying test procedures and assess the impact of delayed testing on recorded G6PD activity and Hb concentration. In addition, the manufacturer-recommended cut-off was evaluated against the site-specific thresholds of two field sites.

## Methods

### Overview

In a Pilot Study several variations of the Biosensor's standard measurement method were trialled at the Menzies School of Health Research, Darwin, Australia. The Standard Method recommended by the manufacturer and the method with the least variability in recorded activity were subsequently assessed at field sites in Malinau Regency, Indonesia, and Kailali District, Nepal. While ethical approval in Indonesia was provided to collect capillary and venous blood from the same participant, Nepalese ethics limited blood collection to either venous or capillary blood. Accordingly, twice as many participants were enrolled in Nepal compared to Indonesia, while the number of measurements performed was the same (Fig 1). Written

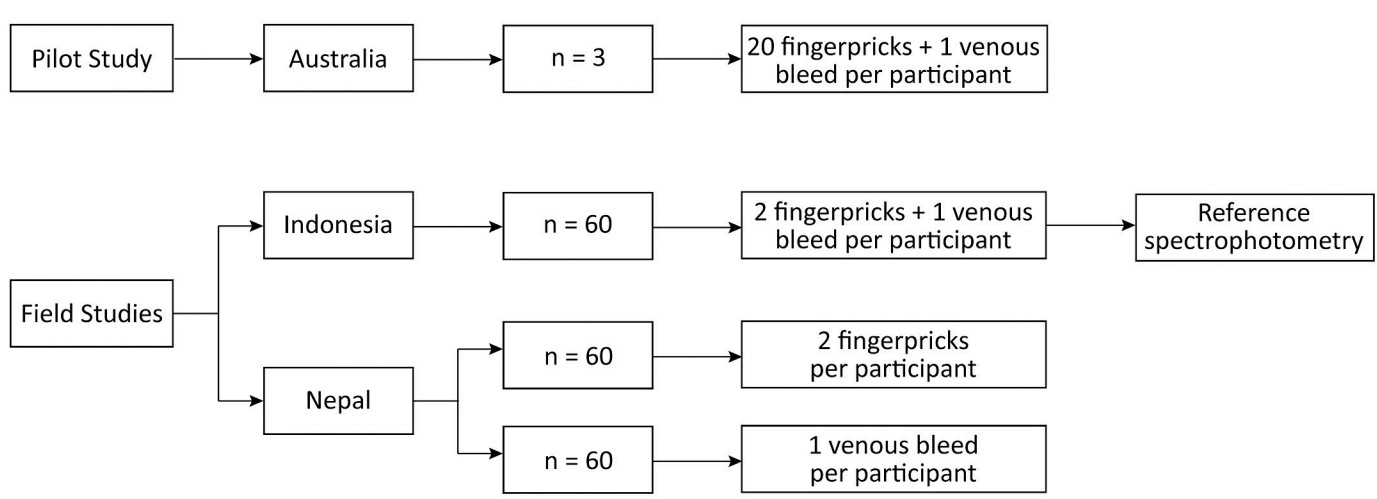

**Fig 1. Schematic workflow of the pilot study in Australia, and the field studies in Indonesia and Nepal.**

informed consent was collected from all participants or their legal guardians prior to enrolment.

## Ethics

Ethical approval was obtained from the Human Research Ethics Committee (HREC) of the Northern Territory Health, Australia (Menzies HREC 22–4346), the Atma Jaya Catholic University Research Ethics Committee, Indonesia (No. 0008A/III/PPPE.PM.10.05/09/2022) and the Institute of Science and Technology, Tribhuvan University Institutional Review Committee, Nepal (No. IRC/IOST60/079/080).

## Assay procedures

- *Standard Method* (recommended by the manufacturer, Fig 2): A single use plastic pipette (Ezi tube, Fig 3) was used to transfer 10 μL of capillary blood from a finger prick to the lysis buffer (included in the test kit) immediately after collection. The blood buffer solution was mixed and 10 μL of the solution was then transferred to the test membrane of a single use test device inserted into the Biosensor. The Biosensor provides normalized G6PD activity (in U/g Hb) and quantifies Hb concentration (in g/dL) within 2 minutes of sample application.

Four variations to the standard method were developed, considering practical relevance in remote field setting, the use of stored blood (capillary and venous) and the use of additional equipment such as micropipettes:

- *Method 1*: Capillary blood was collected in K2 EDTA microtainer tubes (BD, USA; 500 μL) and stored at 4°C; the sample was tested within 60 hours of collection. Prior to testing, the tube was brought to room temperature and 15 μL of blood were transferred from the EDTA tube using a standard micropipette to the surface of a sealing film (Parafilm M, Amcor, USA). Ten μL of blood were collected from the sealing film and testing procedure then followed the standard method.

- *Method 2*: was the same as Method 1, but venous blood instead of capillary blood was collected in a K2 EDTA vacutainer tube (BD, USA; 3mL).

- *Method 3*: was same as Method 2, but the reservoir of the Ezi tube was not squeezed when dispensing the blood buffer solution onto the test membrane of the test device. Instead, the tip of the Ezi tube was placed on the membrane, allowing the buffer blood solution to flow onto the membrane through capillary action.

- *Method 4*: was the same as Method 2, only a calibrated micropipette was used instead of the Ezi tube.

## Pilot study

Variation and repeatability of an assay is relative to the absolute values measured. The lower the absolute "true" value is, the smaller the absolute assay specific variation will be. Considering the inherent background noise of any assay, variation is better measured in samples with higher G6PD activities. The Standard Method and each of the four modified methods were performed in 10 duplicates on three adult participants known not to be G6PD deficient as defined by at least two distinct assays. Twenty finger prick capillary samples (two pricks per finger, 10 μL of capillary blood each, defined as paired samples) were collected from each participant. An additional 400 μL of capillary blood were collected from one of the finger pricks

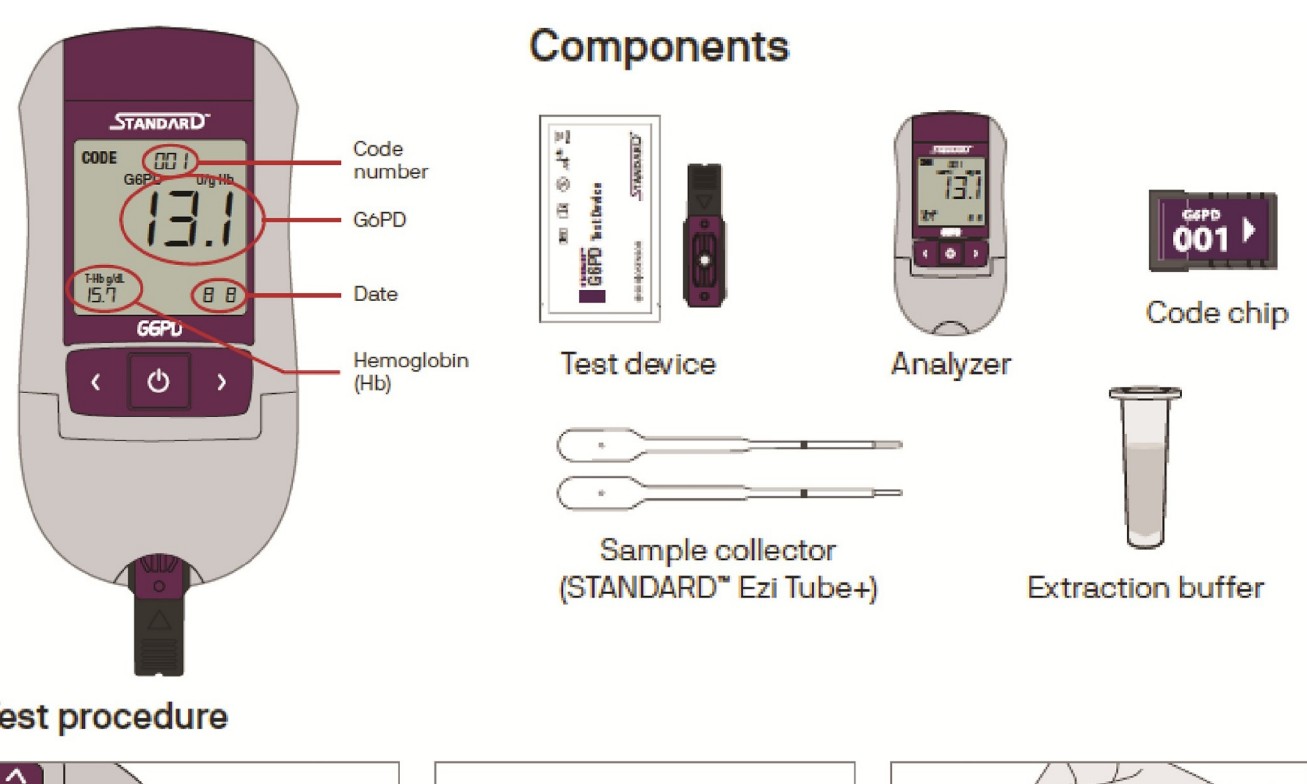

## Test procedure

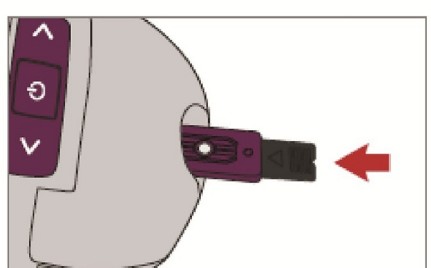

1. Insert test device into analyzer.

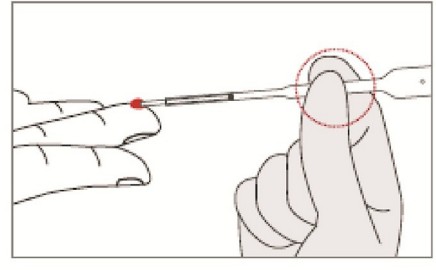

2. Collect blood.

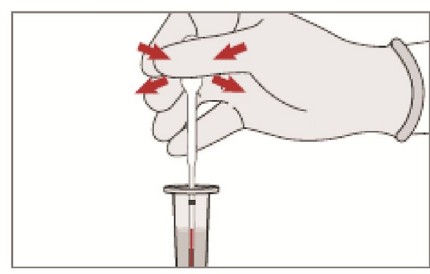

3. Mix blood and buffer 8–10 times.

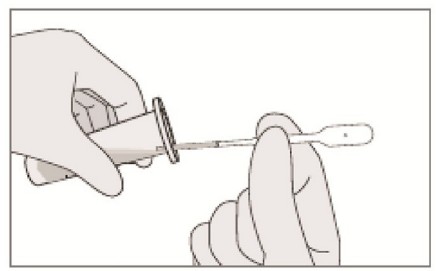

4. Collect mixed sample with NEW sample collector.

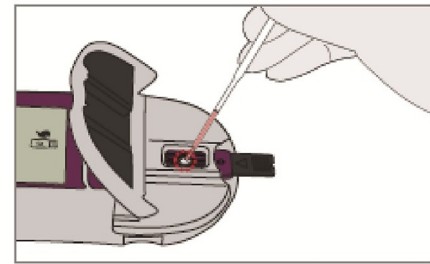

5. Apply the mixed sample to the hole in the test device.

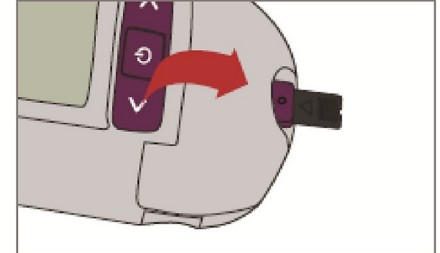

6. Close analyzer flap and wait for 2 minutes.

**Fig 2. Illustrations of components and step-by-step guide of the STANDARD G6PD Test (SD Biosensor, RoK) from Adhikari et al (2022) [29].**

into a K2 EDTA microtainer tube (BD, USA), and 3 mL of venous blood were collected into a K2 EDTA vacutainer tube (BD, USA). Any two consecutive measurements of blood collected from microtainers or vacutainers were defined as paired measurements.

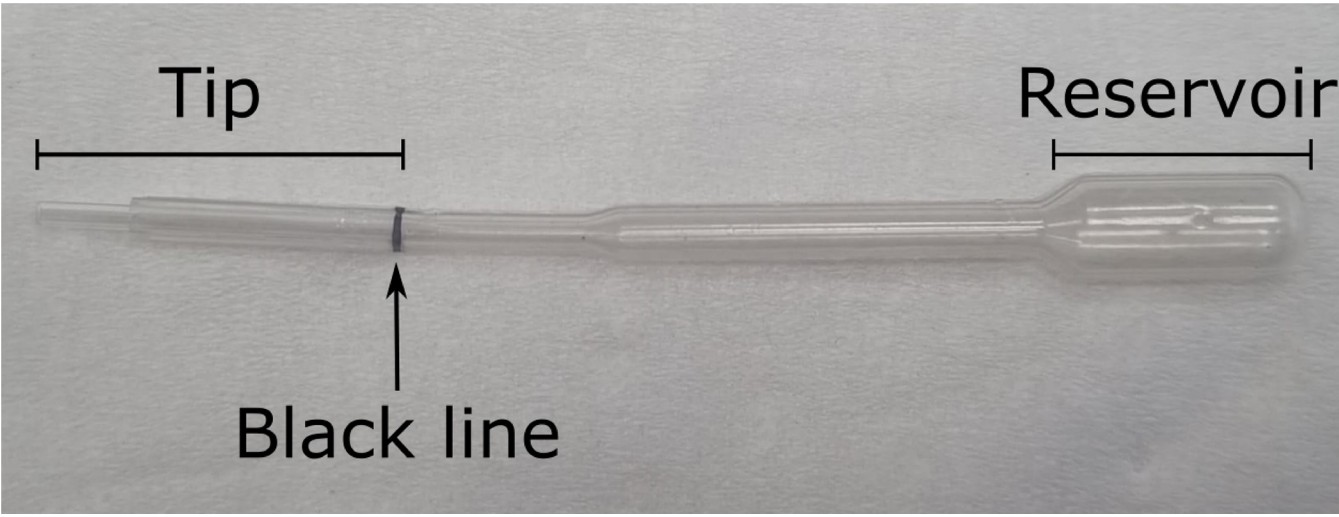

**Fig 3. The Ezi tube, a single use plastic pipette included in the Biosensor test kit.** When the tip is inserted into a blood sample, 10 µL are collected through capillary action. The black line indicates 10µL.

### Field study

In the Indonesian field study two drops of capillary blood (each $\geq$20 µL, defined as paired samples) from two distinct finger pricks were collected from participants above the age of six years (Fig 1). In addition, 3 mL of venous blood were collected from the same participants into a K2 EDTA vacutainer (BD, USA), stored in 4˚C, and tested twice by the biosensor (paired sample). In Nepal the same procedures were followed, however only paired capillary or venous blood samples were collected from each participant, therefore twice as many participants as in Indonesia were enrolled.

### Spectrophotometry (Reference method)

G6PD activity of all Indonesian participants was measured in duplicate from venous blood stored in EDTA tubes by reference spectrophotometry using a commercial kit (Cat. No. G7583-180, Pointe Scientific, USA), executed at 37˚C, read at 340 nm (Biowave II UV/Vis, Biochrom WPA, UK), following kit manufacturer instructions [30]. Ten µL of venous blood were mixed thoroughly with 1 mL of R1 reagent that had been reconstituted with lyse reagent (Cat. No. G7583-LYS). After incubation at room temperature for 5 minutes, 2 mL of R2 reagent were added. The absorbance of the mixture was read at 340 nm after incubation for 5 minutes at 37˚C, and again after another 5-minute incubation at the same temperature. G6PD activity was calculated from the difference of measured absorbance and normalized by Hb reading using the formula provided by the manufacturer. Hb levels were measured with a Hb 301 device (Hemocue, USA) done at the same time as reference testing [31]. G6PD deficient, intermediate, and normal controls (Cat. Nos. HC-108DE, HC-108IN, and HC-108, respectively) were measured daily to assure the quality of reference spectrophotometric readings (ACS Analytics, USA).

### Statistical analysis and sample size calculation

All data were recorded on case report forms and digitalized using EpiData version 3.1 (EpiData Association, Denmark). Data were organized and analysed using STATA versions 13, 15, and 17 (Stata Corp, USA) [32].

The adjusted male median (AMM) was calculated per country from all male participants recruited through the prospective health facility-based surveillance, using the results from the Biosensor Standard Method [33].

To compare the repeatability of the Standard Method and the modified methods, the absolute difference between paired measurements was calculated for each method. In the Pilot Study, the median absolute differences for all methods were compared using the Kruskal-Wallis test. In the Field Study, the median absolute differences of the field-assessed methods were compared using the Wilcoxon signed-rank test (matched samples from Indonesia) or the Mann-Whitney U test (unmatched samples from Nepal). The median absolute differences of each field-assessed methods were compared between the two sites using the Mann-Whitney U test (S1 Table).

In Indonesia the correlation between the Biosensor and spectrophotometry readings was quantified by matching the mean of paired Biosensor readings with the mean of duplicate spectrophotometry readings and calculating the Spearman's rank correlation coefficient ($r_s$). The difference in absolute readings was assessed using the Wilcoxon signed-rank test and Bland-Altman plot analysis. The same procedures were followed to assess the correlation and significance of differences between paired readings of the Biosensor Standard Method and modified method. The proportions of deficient individuals were compared between methods using the McNemar's test for correlated proportions. The median absolute differences of Hb measurements were analysed with the same statistical tests as G6PD activity, and the agreements analysed by calculating Pearson's correlation (r), paired Student T-test, and Bland-Altman plot analysis.

The primary objective of this study was to assess the repeatability of different methods. Repeatability was defined as the difference between paired measurements. To identify a minimal and clinically relevant difference of 0.5 U/g Hb with 80% power and 95% confidence, assuming a standard deviation of 1.5 U/g Hb and a minimal correlation coefficient of $r \geq 0.65$ required recruitment of 52 participants per site. Assuming procedural errors in more than 10% of all participants we enrolled 60 participants for each method at each site to allow for a site-specific analysis using a two-sided approach.

## Results

### Pilot study

Three healthy G6PD normal adult volunteers were enrolled into the Pilot Study between the 4th and 10th of August 2022. Repeatability did not differ significantly between the five Biosensor methods (p = 0.381). The median absolute difference between paired measurements observed when using the Standard Method was 0.5 U/g Hb (interquartile range [IQR]: 0.2 to 0.9, total range: 0.0 to 3.1). Method 3 generated the smallest median absolute difference (0.3 U/g Hb, IQR: 0.2 to 0.4, total range: 0.0 to 2.8) (Table 1 and Fig 4). The Standard Method and Method 3 were subsequently moved forward for evaluation under field conditions.

**Table 1. Comparison of median absolute difference between paired measurements per Biosensor method from the pilot study.**

| Method Name | Blood Collection | Sample Application | Median absolute difference between paired measurements (IQR, total range) in U/g Hb |
|---|---|---|---|
| Standard Method | Capillary | Ezi Tube | 0.5 (0.2–0.9, 0.0–3.1) |
| Method 1 | Capillary EDTA | Ezi Tube | 0.3 (0.2–0.8, 0.0–3.9) |
| Method 2 | Venous EDTA | Ezi Tube | 0.5 (0.2–0.7, 0.0–3.7) |
| Method 3 | Venous EDTA | Ezi Tube (not squeezed) | 0.3 (0.2–0.4, 0.0–2.8) |
| Method 4 | Venous EDTA | Micropipette | 0.5 (0.1–0.9, 0.0–4.5) |

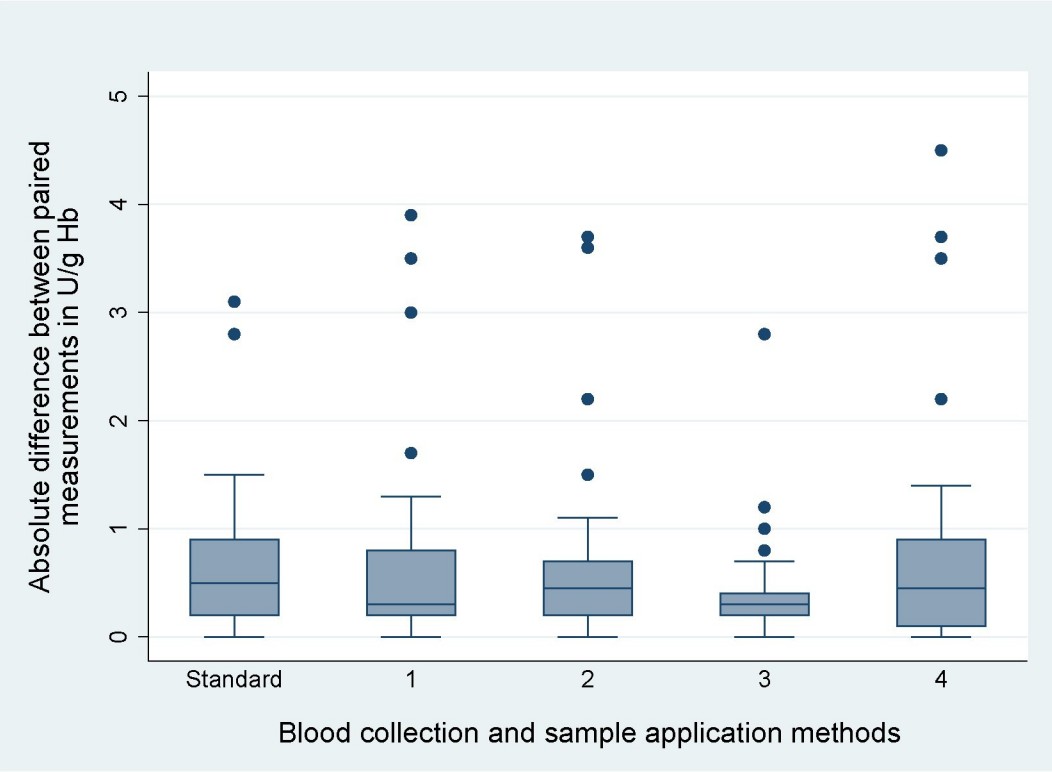

**Fig 4. Absolute differences between paired measurements in U/g Hb for each Biosensor method used in the pilot study.**

### Field study

**Baseline participant data.** In Indonesia 60 participants were enrolled between the 24th and 28th of November 2022, in Nepal 120 participants were enrolled between the 25th of January and 13th of March 2023 (S1 Data). Site-specific AMM by Biosensor Standard Method was calculated to be 6.1 U/g Hb (IQR: 5.3 to 7.5) in Indonesia and 6.1 U/g Hb (IQR: 5.3 to 7.1) in Nepal (Fig 5). None of the Indonesian participants and only 1 (0.8%) Nepalese participant was G6PD deficient (<30% AMM) according to the Biosensor Standard Method. One (1.7%) and 10 (8.3%) participants had intermediate G6PD activity (30–70% AMM) in Indonesia and Nepal, respectively (Table 2).

**Biosensor repeatability of G6PD measurements.** Repeatability of measurements by the Standard Method and Method 3 did not differ in Indonesia (p = 0.425) and Nepal (p = 0.330); Table 3 and Fig 6. The median absolute differences of the Standard Method (0.3 U/g Hb, IQR: 0.1 to 0.5) and Method 3 (0.3 U/g Hb, IQR: 0.1 to 0.6) were similar when combining results from both countries (p = 0.713); Table 3 and Fig 6.

The repeatability of the Standard Method differed significantly between measurements done in Indonesia (0.2 U/g Hb, IQR: 0.1 to 0.4) and in Nepal (0.4 U/g Hb, IQR: 0.2 to 0.6); p = 0.025; Table 3. However, the repeatability for Method 3 did not differ between countries: 0.3 U/g Hb (IQR: 0.1 to 0.5) in Indonesia and 0.3U/g Hb (IQR: 0.1 to 0.6) in Nepal (p = 0.680); Table 3 and Fig 6.

**Biosensor repeatability of Hb measurements.** The repeatability of Hb readings did not differ in Indonesia (p = 0.262), however, in Nepal the Hb median absolute difference of the Standard Method (0.8 g/dL, IQR:0.3 to 1.3) was significantly higher than that of Method 3 (0.3 g/dL, IQR: 0.2 to 0.9); p = 0.006, S3 Table, S1 Fig.

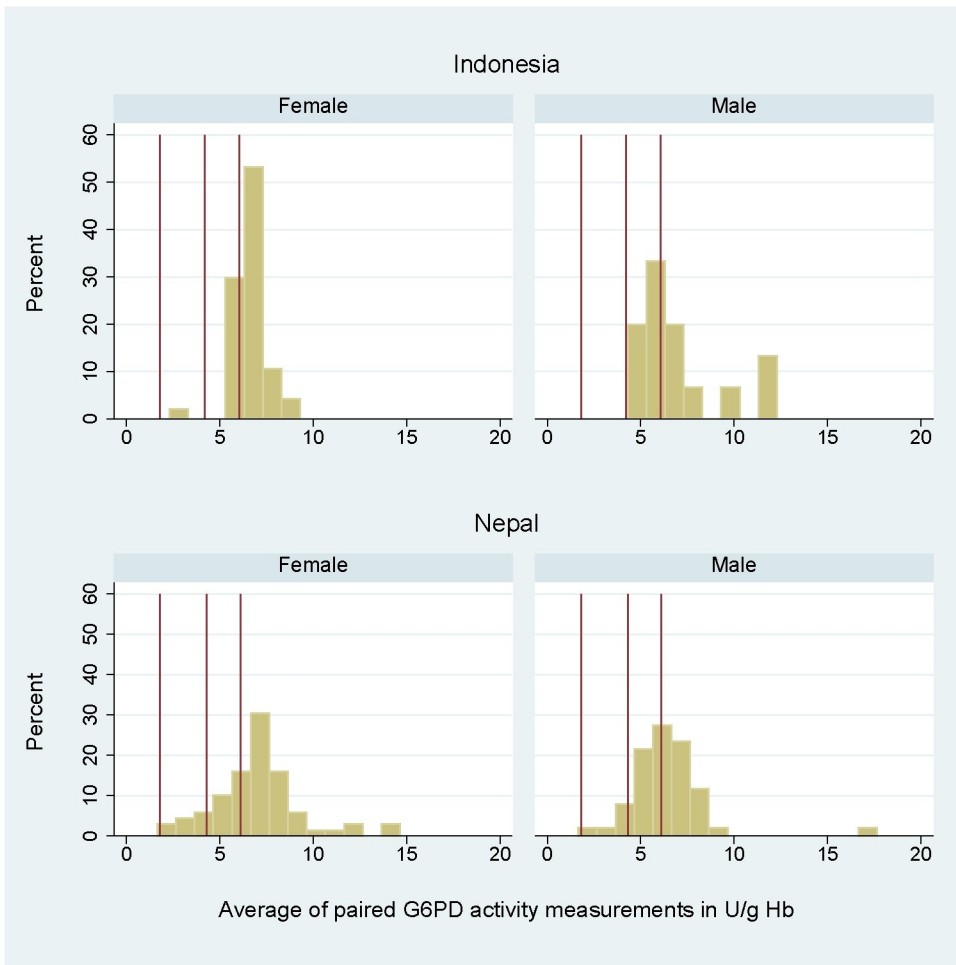

**Fig 5. Histogram of G6PD activity as measured by the Biosensor of female and male participants from Indonesia (Standard Method, n = 60) and Nepal (Standard Method n = 60, and Method 3 n = 60).** Vertical lines denote (from left to right) 30%, 70%, and 100% of AMM.

**Biosensor G6PD readings after time delay and comparison to spectrophotometry.** In Indonesia G6PD activity was measured immediately by the Standard Method, while Method 3 was delayed by a median of 307 minutes (IQR: 275 to 333 minutes) and spectrophotometry by a median of 348 minutes (IQR 133 to 395 minutes); S2 Fig. The median activity by the Standard Method was 6.4 U/g Hb (IQR: 6.0 to 7.0) compared to 6.7 U/g Hb (IQR: 6.2 to 7.3) by Method 3, with a difference of 0.4 U/g Hb (IQR: 0.2 to 0.7); p = 0.005, S3 Fig.

The median activity by spectrophotometry was 10.8 U/g Hb (IQR: 9.3 to 12.2), significantly higher than that derived by the Standard Method (p<0.001) and Method 3 (p<0.001). The correlation between both Biosensor methods ($r_s$ = 0.795; p<0.001) was better than that between spectrophotometry and the Biosensor Standard Method ($r_s$ = 0.451; p<0.001).

Each observation was categorised as deficient (<30%), intermediate (30–70%), or normal (>70%). Categories did not differ between Standard Method or Method 3; Fig 7B. However, when the categories derived from the Biosensor were compared to those from spectrophotometry readings (S2 Table), one individual categorized as deficient by spectrophotometry was categorized as intermediate by the Biosensor, and two individuals that were classified as intermediate by spectrophotometry were categorized as normal by the Biosensor (red dots,

**Table 2. Demography of the field study populations.**

|  | Indonesia | Nepal |
|---|---|---|
| Male | 15 | 51 |
| Female | 45 | 69 |
| Total | 60 | 120 |
| Median age in years (range) | 35 (10 to 81)† | 32 (8 to 90) |
| Median Hb in g/dL (IQR)* | 15.9 (14.6 to 17.1) | 14.3 (13.2 to 16.2) |
| AMM (IQR)* | 6.1 (5.3 to 7.5) | 6.1 (5.3 to 7.1) |
| **G6PD Deficient (Activity <30% AMM)*** | | |
| Male (%) | 0 | 1 (2.0%) |
| Female (%) | 0 | 0 |
| **G6PD Intermediate (Activity 30–70% AMM)*** | | |
| Male (%) | 0 | 4 (7.8%) |
| Female (%) | 1 (2.2%) | 6 (8.7%) |
| **G6PD Normal (Activity >70% AMM)*** | | |
| Male (%) | 15 (100.0%) | 46 (90.2%) |
| Female (%) | 44 (97.8%) | 63 (91.3%) |

†Age data from 59/60 participants.
*Based on the Biosensor Standard Method

Fig 7A). Overall, there was no significant difference between the proportions of deficient and intermediate individuals categorized by the Biosensor vs spectrophotometry (p = 0.317 and p = 0.157 respectively).

**Biosensor vs Hemocue Hb readings.** Matched Hb readings from Biosensor (Standard Method and Method 3) and Hemocue were only assessed in Indonesia (S4 Fig). The mean Hb concentration by the Biosensor Standard Method was 15.9 g/dL (95%CI: 15.4 to 16.3) compared to 16.6 g/dL (95%CI: 16.2 to 17.0) by the Biosensor Method 3 (p<0.001). Both Biosensor measurements were significantly higher than the paired readings of the Hemocue (mean: 13.5 g/dL, 95%CI: 13.2 to 13.9); p<0.001.

## Discussion

Our study demonstrated that the repeatability of the Biosensor was consistent irrespective of type of blood (capillary or venous) and the sample application method used. In the pilot study, Method 3 was the method with the least variability and therefore selected for field assessment. In the field study, the repeatability of G6PD activity measurements did not vary between methods at either site. When repeatability of G6PD activity measurements of either method were compared between sites, there was no statistically significant difference for Method 3; while

**Table 3. Comparison of median absolute difference between paired Biosensor G6PD activity measurements between methods and sites.**

|  | Median absolute difference between paired measurements (IQR) in U/g Hb | | |
|---|---|---|---|
|  | Standard Method | Method 3 | p-value (Standard Method vs Method 3) |
| Indonesia | 0.2 (0.1 to 0.4) | 0.3 (0.1 to 0.5) | 0.425 |
| Nepal | 0.4 (0.2 to 0.6) | 0.3 (0.1 to 0.6) | 0.330 |
| Indonesia + Nepal | 0.3 (0.1 to 0.5) | 0.3 (0.1 to 0.6) | 0.713 |
| p-value (Indonesia vs Nepal) | 0.025 | 0.680 | |

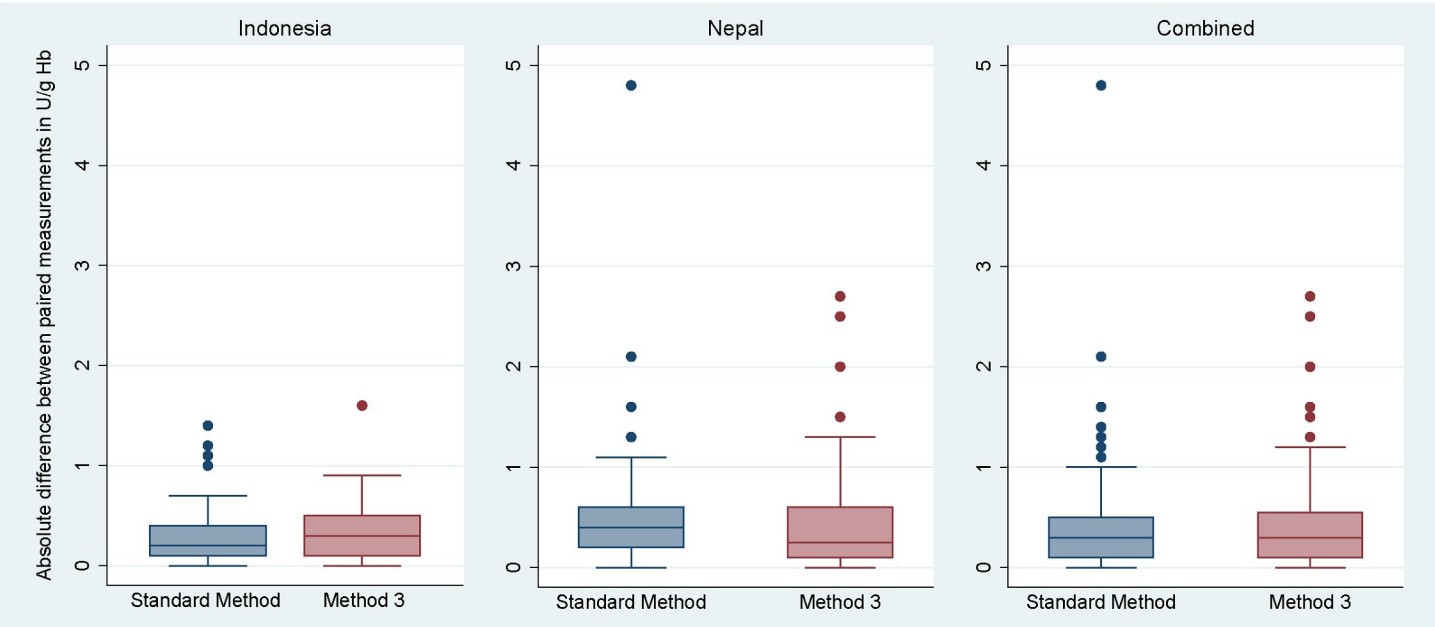

**Fig 6.** Boxplot of absolute differences between paired Biosensor G6PD activity readings per method (in U/g Hb) from field studies in Indonesia (left), Nepal (middle), and both sites (right).

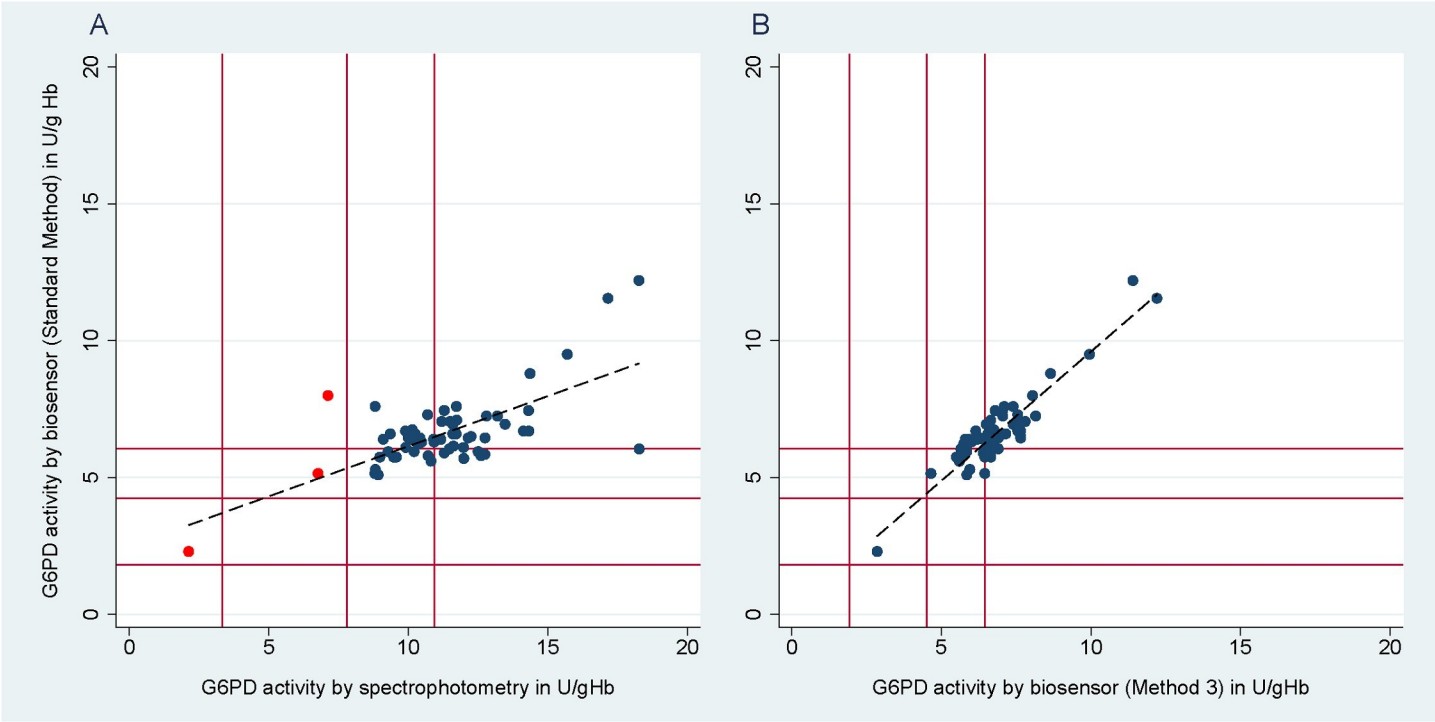

**Fig 7.** Scatterplot of G6PD activity as measured by the Biosensor using the Standard Method vs spectrophotometry (left); and by the Biosensor using the Standard Method vs Method 3 (right). The diagonal line denotes the line of equality. The red horizontal and vertical lines, from the point of origin outwards, mark the 30%, 70%, and 100% of the AMM (normal G6PD activity). Individuals whose G6PD categorization changed when measured by the Biosensor and spectrophotometry were marked as red dots.

the observed difference for the Standard Method was statistically significant, the observed difference is unlikely to be clinically relevant (Table 3). The repeatability of Hb readings from capillary samples showed significantly greater variation compared to venous samples in Nepal but not in Indonesia (S1 Table and S1 Fig), however, this variation did not impact on the repeatability of the G6PD activity.

In the Indonesian field study, G6PD activity by the Standard Method was significantly lower compared to the paired reading of Method 3 performed more than five hours later. However, the median difference was less than 0.5 U/g Hb and thus unlikely to be of clinical relevance.

The readings of both Biosensor methods were well correlated, while the correlation between spectrophotometry and the Biosensor was lower, irrespective of the applied method. Since repeatability and the definition of 100% activity in Indonesia and Nepal were very similar this suggests a degree of imprecision with the reference method and underlines the level of standardization the Biosensor offers irrespective of end user [25, 34].

The AMM was calculated and defined as site-specific 100% G6PD activity based on results from the Biosensor Standard Method [33]. In both countries 100% activity was calculated as 6.1 U/g Hb, an activity that corresponds to the current cut-off for 70% activity which the manufacturer recommends to define the upper limit of intermediate enzyme activity. A recent study from Cambodia reports a site-specific AMM of 6.4 U/g Hb, very close to the manufacturer's 70% cut-off and almost matching our observations [35]. If this observation is confirmed in other settings, this could suggest that the current recommended thresholds may be too conservative.

In the Indonesian cohort, the concentration of Hb measured by the Biosensor was compared to that measured by the Hemocue. The mean differences were 2.3 g/dL and 3.1 g/dL respectively for the Standard Method and Method 3 compared to the Hemocue, with the Biosensor generating the higher readings. A Bland-Altman plot analysis showed that this difference was consistent and not due to extreme outliers (S5 Fig). Past evaluations of the Biosensor have shown differences of less than 0.4 g/dL when compared to the Hemocue Hb201+ [36] or CBC [21, 22]. If the now observed discrepancy is confirmed in other settings, a re-evaluation of the Biosensor Hb readings against a more robust reference may be needed.

Our study has a number of limitations. Only one laboratory and two field sites were involved, with one operator and one machine per site, a design that does not allow to determine the impact of end-users on performance and repeatability. Our study also used different sample application methods on different blood source types, thus the effects of method and blood source on repeatability could not be separated.

In conclusion, there was no significant advantage of modifying the current recommended test procedures, nor was there any difference in readings between capillary and venous blood sampling. The currently recommended cut-off activities to define normal G6PD activities might be conservative, and may lead to misclassification of a proportion of G6PD normal individuals as having intermediate activity. If confirmed in other settings, patients eligible for effective radical cure might be withheld the right treatment. Whilst the Biosensor has the potential to facilitate universal definitions of G6PD deficiency in remote areas [34], the observed difference in Hb readings between the Biosensor and Hemocue warrants further investigation at other sites.

## Supporting information

**S1 Fig.** Boxplot of absolute difference per paired Biosensor Hb readings per method from field studies in Indonesia (left) and Nepal (right).
(TIF)

**S2 Fig. Boxplot of the time differences (in minutes) between blood collection and G6PD activity measurement for Biosensor Standard Method (capillary blood), Biosensor Method**

3 (venous EDTA blood), and spectrophotometry (venous EDTA blood).
(TIF)

**S3 Fig. Bland-Altman plot comparing G6PD activity readings by the Biosensor Standard Method against the Biosensor Method 3; the green dashed line marks the mean difference and the area shaded in grey depicts the 95% limits of agreement.**
(TIF)

**S4 Fig.** Scatterplot of Hb readings by the Biosensor using the Standard Method vs Hemocue (left, r = 0.895, p<0.001); and by the Biosensor using the Standard Method vs Method 3 (right, r = 0.881, p<0.001).
(TIF)

**S5 Fig.** Bland-Altman plots comparing Hb readings by the Biosensor Standard Method (left) and Method 3 (right), both against Hemocue; the green dashed line marks the mean difference and the areas shaded in grey depicts the 95% limits of agreement.
(TIF)

**S1 Table. Summary of groupings and comparisons to assess the repeatability of G6PD measurements by Biosensor.** Definition of a group: the absolute differences of all paired measurements taken using the same method at one site.
(DOCX)

**S2 Table. Summary of Hb, AMM, and G6PD status of Indonesian study population based on reference spectrophotometry.**
(DOCX)

**S3 Table. Comparison of the median absolute difference between paired Biosensor Hb readings between methods and sites.**
(DOCX)

**S1 Data. Corresponding database.**
(XLSX)

## Acknowledgments

We would like to thank all participants who allowed us to collect blood samples. We are thankful to all staff involved in this study, including Mr. Umesh Bajgain, laboratory technologist at Tikapur Hospital, Tikapur, Kailali and Mr. Janak Raj Joshi, laboratory technologist at Malakheti Hospital, Malakheti, Kailali for their help during entire study period.

## Author Contributions

**Conceptualization:** Arkasha Sadhewa, Benedikt Ley, Ari Winasti Satyagraha.

**Data curation:** Arkasha Sadhewa, Benedikt Ley.

**Formal analysis:** Arkasha Sadhewa, Benedikt Ley, Ari Winasti Satyagraha.

**Funding acquisition:** Benedikt Ley.

**Investigation:** Alina Chaudhary, Lydia V. Panggalo, Angela Rumaseb.

**Project administration:** Arkasha Sadhewa, Megha Raj Banjara, Ari Winasti Satyagraha.

**Supervision:** Arkasha Sadhewa, Komal Raj Rijal, Ari Winasti Satyagraha.

**Writing – original draft:** Arkasha Sadhewa.

**Writing – review & editing:** Arkasha Sadhewa, Alina Chaudhary, Lydia V. Panggalo, Angela Rumaseb, Nabaraj Adhikari, Sanjib Adhikari, Komal Raj Rijal, Megha Raj Banjara, Ric N. Price, Kamala Thriemer, Prakash Ghimire, Benedikt Ley, Ari Winasti Satyagraha.

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
