## [Decision Letter · Decision Letter 0]

24 Nov 2023

PONE-D-23-30837Field assessment of the operating procedures of a semi-quantitative G6PD Biosensor to improve repeatability of routine testingPLOS ONE

 Dear Dr. Sadhewa,

Thankyou for submitting your paper. I have a number of comments and suggestions to complement those of the two reviewers. Most are comparatively minor but I do thin you need to be more critical in the Discussion.

Abstract

Line 34 – suggest you add by Biosensor measured G6PD activity

Line 45 – worth saying that the most patients were G6PD normal

Introduction

Line 58 – this sentence implies that some blood stage drugs have antihypnozoite action. I am not aware of any but please include the name of the drug you think has such an action.

Line 77 – a reference for acute kidney injury is required

Lone 99 – not everyone will know the difference between repeatability and reproducibility. Perhaps, these can be defined in the Methods section.

Methods

The methods are a little difficult to follow and I would suggest that more detail is added to Figure 1 to show exactly what was done in the field trial – which method and timing and how each method was analysed i.e. intra group comparison vs. intergroup comparison.

Was there not also an intragroup comparison in Nepal – you took 2 finger pricks per patients?

Line 163 – there is no information on how patients were selected. Did any of them have malaria or other illnesses?

Line 171 – G6PD activity was measured from a venous sample with variable delay. Was pipetting involved in the spectro measurement? If so, we need to know the details to compare with Method 3.

Line 172 – can we assume this was also at 40C and for how long?

Line – 195 – often with Bland Altman analyses, clinicians predefine limits. The authors may wish to do this for the G6PD activity and haemoglobin concentrations. Also, it is conventional to add 95% confidence intervals around the mean difference on the Figures. Can this be done, please?

Results

Line 255 – you mean compared to the HemoCue 301?

Line 263 – with this large difference between the spectrophotometric result and the Biosensor, to me this justifies a Bland Altman analysis between the spectro and standard method with the Figure in the main paper.

Line 270 – I would suggest the G6PD results are integrated in Table 2.

Discussion

Line 306 – did you assess the AMM with the spectrophotometric method?

Line 310 – We need more details about how the manufacturer arrived at higher AMM to justify the statement that their cut off choice is conservative.

Line 315 – so which method is correct? In previous evaluations did the Biosensor also have high Hbs? A mean of 16 g/dL seems quite high which is why it is important to know who was recruited and from where.

Y axis of S5 needs units - g/dL.

Line 317 – does this sentence colour the conclusion in line 310 regarding the SD Biosensor cut off being conservative?

323 – precision. To me this is the same as repeatability. Suggest one or other term is used as the non-specialist may not appreciate that they are the same.

Line 329 – clinical implications?

Line 331 – HemoCue is not a reference method

Limitations

This was quite small study. A full blood count, the reference method for measuring the haemoglobin concentration, was not done. Whilst not essential, genotyping would have been very useful. Most patients were G6PD normal. Applicability to G6PDd and heterozygous females?

General points

The Discussion needs to suggest why we see large differences between the Biosensor and the specto reading and the Biosensor and HemoCue for the Hb. What are the clinical implications of these findings? Are we going to give tafenoquine to the wrong patient?

Ideas for future research?

What about the Biosensor – fit for purpose?

As per one of the reviewers, please critically compare and contrast your findings with the Cambodian study – Adhikari et al. 2022.

We look forward to receiving your revised manuscript.

Kind regards,

Walter RJ Taylor

Academic Editor

PLOS ONE

2. Authors conducting research in other countries or with Indigenous populations are required to complete a copy of PLOS’ questionnaire on inclusivity in global research. The policy applies to researchers who have travelled to a different country to conduct research, research with Indigenous populations or their lands, and research on cultural artefacts. You can find more information on this policy here: https://journals.plos.org/plosone/s/best-practices-in-research-reporting. Unless you choose to reject the manuscript, we will include a request for the authors to complete the questionnaire in the first decision letter. PLOS ONE does not have specific requirements for the responses to questionnaire, beyond our normal ethical standards. However, when the revised manuscript is resubmitted, we would be grateful if you could assess the suitability of the responses and consider whether the research meets all applicable standards for the ethics of experimentation and research integrity. If you wish to discuss the authors’ responses to questionnaire items, please contact us at plosone@plos.org

“This work has received grant funding from the Australia-Indonesia Institute of the Department of Foreign Affairs and Trade of Australia (AII202100069) awarded to BL. AS is supported by Charles Darwin International PhD Scholarships (CDIPS). Publication costs were funded in part by the Division of Teaching of the Menzies School of Health Research”

4. We note that Figure 2 in your submission contain copyrighted images. All PLOS content is published under the Creative Commons Attribution License (CC BY 4.0), which means that the manuscript, images, and Supporting Information files will be freely available online, and any third party is permitted to access, download, copy, distribute, and use these materials in any way, even commercially, with proper attribution. For more information, see our copyright guidelines: http://journals.plos.org/plosone/s/licenses-and-copyright.

Reviewers' comments:

Reviewer's Responses to Questions

**Comments to the Author**

1. Is the manuscript technically sound, and do the data support the conclusions?

Reviewer #1: Yes

Reviewer #2: Yes

2. Has the statistical analysis been performed appropriately and rigorously? 

Reviewer #1: Yes

Reviewer #2: Yes

3. Have the authors made all data underlying the findings in their manuscript fully available?

Reviewer #1: Yes

Reviewer #2: Yes

4. Is the manuscript presented in an intelligible fashion and written in standard English?

Reviewer #1: Yes

Reviewer #2: Yes

5. Review Comments to the Author

Reviewer #1: Field assessment of the operating procedures of a semi-quantitative G6PD Biosensor to improve repeatability of routine testing—is an important study and offers valuable insights on the scope of Biosensor comparing the findings from two countries with a reference method: Spectrophotometry. The article is well written. I have few comments below for authors’ consideration.

On methods: Could the study have recruited a more heterogenous group of participants based on their G6PD status so that it could have generated the measurement for various categories including repeatability?

Also add an explanation on why Indonesia had only 60 participants compared to 120 in Nepal?

Statistical analysis

Explain why data were analyzed using different versions of STATA.

Discussion

Line 295: More explanation needed on this. Does this imply Hb measurement by biosensors are more variable and is not suitable to rely on?

Line 298: Add if it was statistically significant?

Line 301 to 305: Does it imply the limitations of the reference method?

Line 306-310: The discrepancy of AMM based categories and manufacturer's recommendations have been previously shown in Cambodia. Suggest discussing the previous findings.

Line 328-331: Suggest adding with the more recent study from Cambodia that showed discrepancy in manufacturer's recommendation.

In conclusion, authors have well summarized the findings with implications for future studies. But there is also a need to improve the currently deployed SD biosensor owing to some of the user identified shortcomings of the machine highlighted in Cambodian field studies. This also means newer versions could improve these limitations (this may have been overlooked or the manufacturer may have been complacent about it) including the need for more competition from different manufacturers.

Minor:

Figure 3 legend: thorough? Should it be through?

Reviewer #2: This study is interesting and explore different techniques and result of biosensor, and it was also interesting to compare Hb result between biosensor and hemocue. However, there are some questions that need to address :

1. This study used 4 modification techniques but did not mentioned the background of using those different methods, kindly explain.

2. no G6PD deficient sample included in pilot study and only 1 G6PD def samples included in field study, will the result be different if G6PD deficient samples included? please justify why the author only use G6PD normal samples in pilot study.

3. Why did the author use hemocue for normalized Hb reading from spectrophotometry? is this the standard one? please justify and add information about this.

6. PLOS authors have the option to publish the peer review history of their article (what does this mean?). If published, this will include your full peer review and any attached files.

Reviewer #1: No

Reviewer #2: No

---

## [Author Response · Author response to Decision Letter 0]

10 Dec 2023

Author’s note: The lines cited in the responses below refer to the clean version of the revised manuscript (Manuscript.docx).

Response to comments from the Editor

Abstract

Point 1. Line 34 – suggest you add by Biosensor measured G6PD activity

Response 1. We have clarified that it was measured by the Biosensor in the abstract (line 33: “The adjusted male median (AMM) of the Biosensor Standard Method readings was defined as 100% activity.”

Point 2. Line 45 – worth saying that the most patients were G6PD normal

Response 2. We have added this information as follows (lines 37-38): “One Nepalese participant had <30% activity, one Indonesian and 10 Nepalese participants had intermediate activity (≥30% to <70% activity).” 

Introduction

Point 3. Line 58 – this sentence implies that some blood stage drugs have antihypnozoite action. I am not aware of any but please include the name of the drug you think has such an action.

Response 3. We have changed the respective sentence as follows (line 58-59): “Since schizontocidal antimalarial drugs have no activity against hypnozoites, radical cure requires a combination of schizontocidal and hypnozoitocidal agents.”

Point 4. Line 77 – a reference for acute kidney injury is required

Response 4. We have added a respective reference (reference 16, line 77).

Point 5. Line 99 – not everyone will know the difference between repeatability and reproducibility. Perhaps, these can be defined in the Methods section.

Response 5. We have added definitions in the objectives (lines 99-101): “The objective of this study was to assess whether the repeatability (variation of measurement results if the same sample is tested repeatedly under the same conditions) and reproducibility (variation of measurement results if the same sample is tested repeatedly under different conditions)…”

Methods 

Point 6. The methods are a little difficult to follow and I would suggest that more detail is added to Figure 1 to show exactly what was done in the field trial – which method and timing and how each method was analysed i.e. intra group comparison vs. intergroup comparison.

Response 6. We have added a supplementary table (S1_Table.docx) to clarify the groupings and comparisons of the absolute differences of the paired G6PD activity measurements, referred to at line 208 in the text.

Point 7. Was there not also an intragroup comparison in Nepal – you took 2 finger pricks per patients?

Response 7. No, the G6PD measurements done from the two fingerpricks were paired measurements. The absolute difference between these pairs was compared between methods.

Point 8. Line 163 – there is no information on how patients were selected. Did any of them have malaria or other illnesses?

Response 8. The aim of this study was to assess the Biosensor’s repeatability by comparing the difference between paired measurements. Any underlying condition of the patient was not relevant for this analysis. Accordingly, we did not collect information regarding malaria or other illnesses.

Point 9. Line 171 – G6PD activity was measured from a venous sample with variable delay. Was pipetting involved in the spectro measurement? If so, we need to know the details to compare with Method 3.

Response 9. Yes, there were multiple pipetting steps involved. We have added further details on the spectrophotometry procedures to the methods section (lines 185-191): “Ten µL of venous blood were mixed thoroughly with 1 mL of R1 reagent that had been reconstituted with lyse reagent (Cat. No. G7583-LYS). After incubation at room temperature for 5 minutes, 2 mL of R2 reagent were added. The absorbance of the mixture was read at 340 nm after incubation for 5 minutes at 37 °C, and again after another 5-minute incubation at the same temperature. G6PD activity was calculated from the difference of measured absorbance and normalized by Hb reading using the formula provided by the manufacturer. Hb levels were measured with a Hb 301 device (Hemocue, USA) done at the same time as reference testing.”

Point 10. Line 172 – can we assume this was also at 40C and for how long?

Response 10. Incubation temperature was at 37 °C for 5 minutes prior to the first measurement, then another 5 minutes before measuring for the second time. We have altered this section accordingly (kjndly see response 9 and lines 187-189): “The absorbance of the mixture was read at 340 nm after incubation for 5 minutes at 37 °C, and again after another 5-minute incubation at the same temperature.”

Point 11. Line 195 – often with Bland Altman analyses, clinicians predefine limits. The authors may wish to do this for the G6PD activity and haemoglobin concentrations. Also, it is conventional to add 95% confidence intervals around the mean difference on the Figures. Can this be done, please?

Response 11. We have added in the captions of Figures S3 and S5 that the grey shaded areas are 95% limits of agreement (lines 502-504 and 510-512, respectively). This was an exploratory study and accordingly, we did not predefine any limits of difference for the Bland-Altman analyses. 

Results

Point 12. Line 255 – you mean compared to the HemoCue 301?

Response 12. (Now lines 278-280): This sentence refers to repeatability and does not involve a comparison with a reference method. To assess repeatability, we compared paired Hb readings within the Standard Method and Method 3. We explain how we calculated repeatability within the methods (lines 202-203): “To compare the repeatability of the Standard Method and the modified methods, the absolute difference between paired measurements was calculated for each method.” In the results section we again emphasize that we talk about repeatability (lines 278-280): “The repeatability of Hb readings did not differ in Indonesia (p=0.262), however, in Nepal the Hb median absolute difference of the Standard Method (0.8 g/dL, IQR:0.3 to 1.3) was significantly greater than that of Method 3 (0.3 g/dL, IQR: 0.2 to 0.9); p=0.006, S1 Table, S1 Fig.” .

Point 13. Line 263 – with this large difference between the spectrophotometric result and the Biosensor, to me this justifies a Bland Altman analysis between the spectro and standard method with the Figure in the main paper.

Response 13. (Now line 286) We included a scatterplot with 30, 70, and 100% AMM lines (Figure 7) to illustrate correlation and categorization agreement instead of a Bland-Altman analysis. A direct comparison of different assays is not informative as has been demonstrated by Pfeffer et al [1]. 

Point 14. Line 270 – I would suggest the G6PD results are integrated in Table 2.

Response 14. We have added the spectrophotometry results, which were only done in Indonesia, in supplementary Table S2, referred to in line 295 in the text. 

Discussion

Point 15. Line 306 – did you assess the AMM with the spectrophotometric method?

Response 15. Yes, we did. Kindly see the recently added Table S2. Kindly see our reply to Point 13 above, given that absolute values for G6PD activity differ by assay, a direct comparison of AMM by spectrophotometry and Biosensor is not informative.

Point 16. Line 310 – We need more details about how the manufacturer arrived at higher AMM to justify the statement that their cut off choice is conservative.

Response 16. We do not know how the manufacturer arrived at the suggested definitions. We had enquired with the manufacturer but did not get a clear reply. We agree that this point requires clarification but think this is beyond the scope of this article. We do however report that the site-specific cut-offs we calculated for Indonesia and Nepal suggest fairly conservative definitions by the manufacturer.

Point 17. Line 315 – so which method is correct? In previous evaluations did the Biosensor also have high Hbs? A mean of 16 g/dL seems quite high which is why it is important to know who was recruited and from where.

Response 17. (Now line 342) The Hemocue is probably the most widely used point-of-care device to measure haemoglobin levels, but it is not the reference method (this is complete blood count (CBC)). Previous Biosensor evaluations were cited in line 343 and did not show a significant difference between Hb measurement by Biosensor and Hemocue or CBC. However, these evaluations did not include absolute numbers of Hb levels, only comparisons in the form of Bland-Altman analyses. We agree that a mean of 15.9 g/dL in this study is quite high and is significantly higher than the mean as measured by Hemocue. This will require additional follow up as we have clarified in the conclusion (lines 358-359): “…the observed difference in Hb readings between the Biosensor and Hemocue warrants further investigation…”

Point 18. Y axis of S5 needs units - g/dL.

Response 18. We have revised Figure S5 accordingly.

Point 19. Line 317 – does this sentence colour the conclusion in line 310 regarding the SD Biosensor cut off being conservative?

Response 19. (Now line 343) Following discussion among the authors we have realized that the study design did not allow us to distinguish between a machine inherent erroneous Hb reading and an end user error as stated in the limitations lines 347-349. We have therefore removed the respective sentence.

Point 20. Line 323 – precision. To me this is the same as repeatability. Suggest one or other term is used as the non-specialist may not appreciate that they are the same.

Response 20. (Now line 349) We have replaced the word “precision” with “repeatability”.

Point 21. Line 329 – clinical implications?

Response 21. We added a possible implication of the manufacturer-recommended cut-off being too conservative (lines 356-357): “If confirmed in other settings, patients eligible for effective radical cure might be withheld the right treatment.” 

Point 22. Line 331 – HemoCue is not a reference method

Response 22. (Now lines 359). We replaced “the reference method” with “Hemocue”.

Limitations

Point 23. This was quite small study. A full blood count, the reference method for measuring the haemoglobin concentration, was not done. Whilst not essential, genotyping would have been very useful. Most patients were G6PD normal. Applicability to G6PDd and heterozygous females?

Response 23. We have added our sample size considerations to the methods (lines 219-225): “The primary objective of this study was to assess the repeatability of different methods. Repeatability was defined as the difference between paired measurements. To identify a minimal and clinically relevant difference of 0.5 U/g Hb with 80% power and 95% confidence, assuming a standard deviation of 1.5 U/g Hb and a minimal correlation coefficient of r≥0.65 required recruitment of 52 participants per site. Assuming procedural errors in more than 10% of all participants we enrolled 60 participants for each method at each site to allow for a site-specific analysis using a two-sided approach.” We have justified the large proportion of phenotypically G6PD normal participants (lines 162-165): “Variation and repeatability of an assay is relative to the absolute values measured. The lower the absolute “true” value is, the smaller the assay specific absolute variation will be. Considering the inherent background noise of any assay, variation is better measured in samples with higher G6PD activities.” 

General points

Point 24. The Discussion needs to suggest why we see large differences between the Biosensor and the specto reading and the Biosensor and HemoCue for the Hb. What are the clinical implications of these findings? Are we going to give tafenoquine to the wrong patient?

Response 24. Whether the evaluated assay is suitable to guide tafenoquine based radical cure is best answered by assessing the device’s performance (sensitivity and specificity). This has been done repeatedly [2] and was not part of this study. In this study we assessed whether we could improve repeatability, highly relevant when introducing the device into routine care. We therefore do not comment on the assay suitability to guide radical cure.

Point 25. Ideas for future research?

Response 25. Plenty! Amongst others we conclude with the following statement (lines 357-359): “Whilst the Biosensor has the potential to facilitate universal definitions of G6PD deficiency in remote areas [33], the observed difference in Hb readings between the Biosensor and the Hemocue warrants further investigation at other sites,” 

Point 26. What about the Biosensor – fit for purpose?

Response 26. Our study focused on several aspects of the Biosensor. As stated in our concluding paragraph (lines 352-359), we found that the Biosensor gave consistent results regardless of procedure modifications, blood type, or time delay. However, the concerns with the manufacturer-recommended cut-off and the observed high Hb readings warrant further investigations.

Point 27. As per one of the reviewers, please critically compare and contrast your findings with the Cambodian study – Adhikari et al. 2022.

Response 27. We have added a section where we compare our findings to those of Adhikari et al (lines 333-336): “A recent study from Cambodia report a site-specific AMM of 6.4 U/g Hb, very close to the manufacturer’s 70% cut-off [34]. If this observation is confirmed in other settings, it suggests that the current recommended threshold may be too conservative.” 

Response to comments from Reviewer 1

Point 1. Could the study have recruited a more heterogenous group of participants based on their G6PD status so that it could have generated the measurement for various categories including repeatability?

Response 1. Variation and repeatability in G6PD measurement are relative to the measured absolute G6PD activity. In individuals with lower G6PD activity, the observed variation will be smaller and harder to distinguish from the inherent background noise of the assay. In line with our aim to assess repeatability, we only recruited G6PD normal individuals for the pilot study and did not use any G6PD activity based inclusion criteria in the field studies. We have added this explanation to the Methods section, lines 162-165: “Variation and repeatability of an assay is relative to the absolute values measured. The lower the absolute “true” value is, the smaller the absolute assay specific variation will be. Considering the inherent background noise of any assay, variation is better measured in samples with higher G6PD activity.”

Point 2. Also add an explanation on why Indonesia had only 60 participants compared to 120 in Nepal?

Response 2 We have added the following section at the beginning of the methods (lines 112-116): “While ethical approval in Indonesia was provided to collect capillary and venous blood from the same participant, Nepalese ethics limited blood collection to either venous or capillary blood. Accordingly, twice as many participants were enrolled in Nepal compared to Indonesia, while the number of measurements performed was the same (Fig. 1).” 

Point 3. Explain why data were analyzed using different versions of STATA.

Response 3. Different versions of STATA were used between the author’s personal computer (version 13) and the work computer used (versions 15). There was a centralized update from version 15 to 17 in our institution’s computers while preparing this manuscript.

Point 4. Line 295: More explanation needed on this. Does this imply Hb measurement by biosensors are more variable and is not suitable to rely on?

Response 4. (Now line 321): The sentence “The repeatability of Hb readings from capillary samples showed significantly greater variation compared to venous samples in Nepal but not in Indonesia (S1 Table and S1 Fig.), however, this variation did not impact on the repeatability of the G6PD activity.” refers to comparison of variability between the two Biosensor methods. The implication was that one method was more variable than the other in measuring Hb, but only in the Nepali field study. 

Point 5. Line 298: Add if it was statistically significant?

Response 5. The difference was statistically significant but unlikely of any practical significance. We have clarified this. The revised sentence now reads (lines 324-326): “In the Indonesian field study, G6PD activity by the Standard Method was significantly lower compared to the paired reading of Method 3 performed more than five hours later. However, the median difference was less than 0.5 U/g Hb and thus unlikely to be of clinical relevance.” 

Point 6. Line 301 to 305: Does it imply the limitations of the reference method?

Response 6. (Now lines 327-331) Yes. As explored in Pfeffer et al’s meta-analysis of the reference method [1] it has been shown that G6PD measurements using the reference method vary due to the multiple brands of assay kits and differences in spectrophotometry instruments.

Point 7. Line 306-310: The discrepancy of AMM based categories and manufacturer's recommendations have been previously shown in Cambodia. Suggest discussing the previous findings.

Response 7. We have added a section where we compare our findings to those of Adhikari et al (lines 335-337): “A recent study from Cambodia report a site-specific AMM of 6.4 U/g Hb, very close to the manufacturer’s 70% cut-off and almost matching our observations. If this observation is confirmed in other settings, it suggests that the current recommended threshold may be too conservative.” 

Point 8. Line 328-331: Suggest adding with the more recent study from Cambodia that showed discrepancy in manufacturer's recommendation.

Response 8. We have added comparison in lines 335-337, kindly see response to Point 7 above.

Point 9. Figure 3 legend: thorough? Should it be through?

Response 9. Thank you for pointing out the mistake. It has been revised accordingly (line 159): “When the tip is inserted into a blood sample, 10 µL are collected through capillary action.”

Response to comments from Reviewer 2

Point 1. This study used 4 modification techniques but did not mentioned the background of using those different methods, kindly explain.

Response 1. We have added the explanation to the Methods section, lines 136-138: “Four variations to the standard method were developed, considering practical relevance in remote field setting, including the use of stored blood (capillary and venous) and additional equipment such as micropipettes:”

Point 2. No G6PD deficient sample included in pilot study and only 1 G6PD def samples included in field study, will the result be different if G6PD deficient samples included? please justify why the author only use G6PD normal samples in pilot study.

Response 2. Variation and repeatability in G6PD measurement are relative to the measured absolute G6PD activity. In individuals with lower G6PD activity, the observed variation will be smaller and harder to distinguish from the inherent background noise of the assay. In line with our aim to assess repeatability, we only recruited G6PD normal individuals for the pilot study and did not use any G6PD activity based inclusion criteria in the field studies. We have added this explanation to the Methods section, lines 162-165: “Variation and repeatability of an assay is relative to the absolute values measured. The lower the absolute “true” value is, the smaller the absolute assay specific variation will be. Considering the inherent background noise of any assay, variation is better measured in samples with higher G6PD activities.”

Point 3. Why did the author use hemocue for normalized Hb reading from spectrophotometry? is this the standard one? please justify and add information about this. 

Response 3. The gold standard for Hb measurement is complete blood count (CBC) but the field site in Indonesia was remote and did not have access to a CBC machine. It is common practice to replace CBC by Hemocue instead since the device shows consistent good performance. We have added the following reference to support this argument [3] (reference 31 in the manuscript).

References

1. Pfeffer DA, Ley B, Howes RE, Adu P, Alam MS, Bansil P, et al. Quantification of glucose-6-phosphate dehydrogenase activity by spectrophotometry: A systematic review and meta-analysis. PLOS Medicine. 2020;17(5):e1003084. doi: 10.1371/journal.pmed.1003084.

2. Zobrist S, Brito M, Garbin E, Monteiro WM, Clementino Freitas S, Macedo M, et al. Evaluation of a point-of-care diagnostic to identify glucose-6-phosphate dehydrogenase deficiency in Brazil. PLOS Neglected Tropical Diseases. 2021;15(8):e0009649. doi: 10.1371/journal.pntd.0009649.

3. Jain A, Chowdhury N, Jain S. Intra- and inter-model reliability of Hemocue Hb 201+ and HemoCue Hb 301 devices. Asian J Transfus Sci. 2018;12(2):123-6. doi: 10.4103/ajts.AJTS_119_17. PubMed PMID: 30692796; PubMed Central PMCID: PMCPMC6327767.

---

## [Editor Report · Decision Letter 1]

18 Dec 2023

Field assessment of the operating procedures of a semi-quantitative G6PD Biosensor to improve repeatability of routine testing

PONE-D-23-30837R1

Dear Dr. Sadhewa, 

We’re pleased to inform you that your manuscript has been judged scientifically suitable for publication and will be formally accepted for publication once it meets all outstanding technical requirements.

Kind regards,

Walter RJ Taylor

Academic Editor

PLOS ONE

---

## [Editor Report · Acceptance letter]

10 Jan 2024

PONE-D-23-30837R1 

PLOS ONE

Dear Dr. Sadhewa, 

I'm pleased to inform you that your manuscript has been deemed suitable for publication in PLOS ONE. Congratulations! Your manuscript is now being handed over to our production team.

Kind regards, 

on behalf of

Dr. Walter RJ Taylor 

Academic Editor

PLOS ONE